# Genetic Diversity of Potyviruses Associated with Tulip Breaking Syndrome

**DOI:** 10.3390/plants9121807

**Published:** 2020-12-19

**Authors:** János Ágoston, Asztéria Almási, Katalin Salánki, László Palkovics

**Affiliations:** 1Department of Plant Pathology, Faculty of Horticultural Science, Szent István University, 1118 Budapest, Hungary; agoston.janos@kvk.uni-neumann.hu; 2Department of Agriculture, Faculty of Horticulture and Rural Development, John von Neumann University, 6000 Kecskemét, Hungary; 3Plant Protection Institute, Centre for Agricultural Research, 1022 Budapest, Hungary; almasi.aszteria@atk.hu (A.A.); salanki.katalin@atk.hu (K.S.)

**Keywords:** tulip, *Tulipa*, *Potyvirus*, tulip breaking syndrome, *Tulip breaking virus*, *Lily mottle virus*, Rembrandt tulip-breaking virus, recombination, RDP, Hungary

## Abstract

Tulip breaking is economically the most important viral disease of modern-day tulip growing. It is characterized by irregular flame and feather-like patterns in the flowers and mosaic on the foliage. Thirty-two leaf samples were collected from cultivated tulip plants showing tulip breaking syndrome from Hungary in 2017 and 2018. Virus identification was performed by serological (ELISA) and molecular (RT-PCR) methods. All samples proved to be infected with a potyvirus and evidence was provided that three potyvirus species could be identified in the samples: *Lily mottle virus* (LMoV), *Tulip breaking virus* (TBV) and Rembrandt tulip-breaking virus (ReTBV). Recombination prediction accomplished with Recombination Detection Program (RDP) v4.98 revealed potential intraspecies recombination in the case of TBV and LMoV. Phylogenetic analyses of the coat protein (CP) regions proved the monophyletic origin of these viruses and verified them as three different species according to current International Committee on Taxonomy of Viruses (ICTV) species demarcation criteria. Based on these results, we analyzed taxonomic relations concerning potyviruses associated with tulip breaking syndrome. We propose the elevation of ReTBV to species level, and emergence of two new subgroups in ReTBV.

## 1. Introduction

Tulips, as ornamental plants, have first been described as early as 1576 by Carolus Clusius [1]. Tulips are bulbous ornamental plants belonging to the genus *Tulipa* [2,3]. Currently 102 accepted taxa [4] with more than 6700 cultivars belong to the genus [5]. The vast majority of the cultivars are the hybrids of *Tulipa gesneriana* [2,3,6]. Tulips are vegetatively propagated under open field conditions. Only breeders and plant enthusiasts grow tulips from seed. The juvenile phase is around six years [7,8,9], and it takes about 25 years to bring a new cultivar to the market [8,9]. The vegetative reproduction, to have a uniform and identical crop, and the relatively low vegetative propagation rate, makes this crop susceptible to a variety of diseases, especially to viruses. Tulip breaking is economically the most important viral disease of modern-day tulip growing. In the early years of the 17th century, tulips with broken flowers were highly appreciated, they became a symbol of status [7,10]. Later, these broken tulips were called Rembrandt tulips, and became the second known, but the first well documented record of viral disease of a plant [11]. Tulip breaking syndrome is characterized by irregular flame and feather like patterns in the flowers and line pattern and mosaic on the foliage [12]. Symptoms occur in a short period during the growing season, usually before or during flowering, occasionally after flowering. Newly infected tulips show symptoms only in the next growing season [12,13,14]. In spring when infected plants emerge the leaves stand stiffly upright, become crooked or cigar shaped, and are often smaller than normal. Some cultivars show purple-burgundy or light and dark green striping on the abaxial side and near the leaf margins [13,15]. These symptoms are transient, they fade or become masked as the growing season progresses. At the beginning of flowering or after flowering leaves develop the characteristic striping or ringspot mosaic [13,15,16]. Infected plants go dormant sooner than healthy ones [13]. The pattern, the type of color change, and the contrast in the breaking depends on the cultivar [13,15]. In the case of white, cream, and yellow flowers breaking causes glossy flecks on the petals, which is hard to spot in the field. In the case of cultivars with green stigma infected ones became lighter, and originally light green ones turn white. Red flowered plants may have a red blush on their green styles. In all cases, the color change intensifies as the flowering progresses [13].

Several species of viruses were identified on tulips from different genera like Arabis mosaic virus (Nepovirus), Lily symptomless virus (Carlavirus), Cucumber mosaic virus (Cucumovirus), Potato virus X and Tulip virus X (Potexvirus), Tobacco necrosis virus (Necrovirus), Tobacco rattle virus (Tobravirus), and different members of the family Potyviridae, genus Potyvirus [13,14]. The infection of potyviruses are thought to be the primary reason of tulip breaking [7,10,17].

Previously different potyviruses were identified from tulips, including tulip breaking virus (TBV), tulip band-breaking virus (TBBV), tulip top-breaking virus (TTBV) that is a strain of turnip mosaic virus (TuMV-TTB), Rembrandt tulip-breaking virus (ReTBV), and lily mottle virus (LMoV) [13].

According to the International Committee on Taxonomy of Viruses (ICTV) four tulip infecting potyvirus species are accepted. *Lily mottle virus* (LMoV, Accession number (acc. #): AJ564636, RefSeq: NC_005288) [14,18] was accepted as a species in 2000 [19]. Tulip band breaking virus (TBBV) became a synonym of LMoV because of the high nucleic acid identity according to the fast track proposal FT2003.020P.01. [20], and it is also known as tulip breaking virus lily strain (TBV-lily) [21,22]. The vast majority of the LMoV sequence data in the GenBank originated from different lily species, as it has great importance in lily bulb and flower production.

*Tulip breaking virus* (TBV, acc. #: KF826466, RefSeq: NC_043168) [18,23] was accepted as a species in 1976 [24], the first complete coat protein sequence was published in 1994 (acc. #: X63630) [25]. It is present all over the world primarily on tulip plants.

*Tulip mosaic virus* (TulMV, acc. #: X63630, RefSeq: NC_043425) [18] was identified as a separate species in 2005 [26], but the reference sequence presented in the ICTV database shows 98.78% identity with acc. # KF442403 and 96.96% identity with acc. # MH886517, both are TBV isolates. 

Tulip chlorotic blotch virus [18,23] and Tulip top-breaking virus (TTBV) [14] were merged into *Turnip mosaic virus* (TuMV, acc. #: AF169561, RefSeq: NC_002509) [19] and occurs rarely on tulip. In 2002, TTBV was identified in a tulip field in the Netherlands, next to a broccoli field, and as aphids were present presumable transmission occurred [27]. 

Rembrandt tulip-breaking virus (ReTBV, acc. #: S60808) was accepted as species in 1995 [28], but moved to tentative species in fast track proposal FT2003.021P.01., because only a partial coat protein (CP) nucleotide sequence [20] was available. 

As data on nucleic acid sequence accumulated, the virus species demarcation got stricter and a lot of inconsistencies have been cleared up. In the case of potyviruses ICTV species demarcation criteria requires less than 82% identity of amino acid and less than 76% identity of nucleotide sequence of the polyprotein or CP [29] and adequate difference of the polyprotein cleavage site is also required.

In the present study, we aimed to determine the diversity of potyviruses present in Hungarian tulip plants showing typical color breaking syndrome. We also analyzed the taxonomic relationships between tulip infecting potyviruses and incidence of recombination in the evolution of tulip infecting potyviruses.

## 2. Results

### 2.1. Detection and Identification of Potyviruses in Tulips

Thirty-two tulip samples showing typical tulip breaking symptoms were collected in different locations in Hungary (Figure 1).

All of the collected tulip samples proved to be infected with a potyvirus, according to potyvirus specific ELISA tests (Appendix A). The nucleotide (nt) sequences of the cloned fragments (C terminal region of the nuclear inclusion protein b (NIb), complete coat protein (CP), and the 3’ nontraslated region (3′ NTR) were determined, sequences were annotated and uploaded into GenBank. According to the homology with GenBank data 16 TBV, 13 LMoV, and 3 ReTBV infections were identified. To analyze whether single or multiple potyvirus infection occurred in the tulip samples, all of the original PCR products derived from the tulip plants were digested with virus species specific restriction endonucleases, and in all cases the PCR products were completely digested verifying single virus infection (Appendix A). The potyvirus infected plant samples were depicted on map, to illustrate the spatial distribution (Figure 1). LMoV and TBV were present at diverse locations all over the country, while ReTBV was identified solely in Budapest (Figure 1). Regarding host preference in the Darwin-hybrid group LMoV (9) and TBV (9) were present in equal numbers, but in the Rembrandt group only ReTBV was present, as expected. In a tulip garden in Szada (samples 20–27) and Gödöllő (samples 1 and 2), both LMoV and TBV were present in equal number (4–4 and 1–1). Comparing the symptoms on whole plants viruses could not be separated from one another solely based on symptoms (Figure 2, Figure 3 and Figure 4). 

All virus infected plants showed early symptoms, purple-burgundy stripes, or flecks on the abaxial surface of the leaves early in the season (Figure 2A, Figure 3A and Figure 4A,C), but later these symptoms became masked as leaves unfolded. Symptoms on the expanded leaves were usually blurry light and dark striping (Figure 2B,C). Typical flower breaking symptoms were observed on petals in each case when flowers emerged. Breaking was fleck type on cultivars Barbados (Figure 2E), Crystal Beauty (Figure 2F), Apeldoorn (Figure 2I) and Gudoshnik (Figure 3G), flame or feather like on Groenland (Figure 2D), Lambada (Figure 2G), Oscar (Figure 2L), Apeldoorn (Figure 2H,J,K and Figure 3E,F), Claudia (Figure 3H), Absalon (Figure 4D), Insulinde (Figure 4E), and Zomerschoon (Figure 4F). Sometimes the base color of the flower changed, as in the cases of infected Groenland (Figure 2D)—from pink to reddish pink—and Lambada (Figure 2G)—from pinkish orange to red and yellow. White flowers may not show remarkable flower breaking (Figure 3I). 

### 2.2. Recombination

Recombination Detection Program (RDP) v4.98 predicted six recombination events with at least 3 different methods (Table 1). Each event was predicted only in one sequence, and all events showed intraspecific recombination. The summary of recombination events is presented in Table 2. Arabic numbers indicate the order in which RDP characterized the recombination events. 

The average *p*-values (Bonferroni corrected) were calculated for each recombination event, the highest acceptable *p*-value was 0.05 (Table 2).

Three potential recombination events were predicted in the case of LMoV strains, in one isolate (acc. # JN127341) multiple recombination events were identified. In the case of TBV isolates three potential recombination events were predicted, but no evidence of potential recombination was found in ReTBV isolates. Most of the predicted recombination breakpoints are located in the CP, but recombination points in the NIb protein coding region and in the 3′ UTR were also predicted (Figure 5).

#### 2.2.1. Recombination in LMoV Strains

Three potential recombination events were predicted in the case of LMoV. In an Australian strain (acc. #: JN127341) isolated from *Lilium longiflorum* two recombination events were predicted and one recombination was predicted in a *Lilium* strain of LMoV (acc. #: AJ310203).

In the case of the Australian LMoV strain (acc. #: JN127341), in the first predicted recombination, the major parent is unknown, the program used MK368802 (subgroup I) to infer the potential major parent, minor parent is the LMoV strain from Ohio (acc. #: KF553658, subgroup II). The first breakpoint is at position 645 at the amino-terminal part of the CP, ending breakpoint is undetermined, but probably at position 1398, also located in the CP. Graphic representation of the event and Neighbour Joining (NJ) trees of both parents are shown on Figure 6. The nt sequence derived from the major parent has a special location on the phylogenetic tree, it seems to have a distinct origin as it is characterized as recombinant in event 5, while the fragment derived from the minor partner clearly belongs to LMoV subgroup II. 

In case of the other predicted recombination in the LMoV strain acc. #: JN127341 (5th recombination event) the major parent was identified as LMoV acc. #: AJ310203 (subgroup I)—which was detected as a recombinant in event four—potential minor parent is unknown, the program used LMoV strain MK368802 (subgroup I) to infer it. The beginning breakpoint was at position 78 in the NIb protein coding region, while the second breakpoint was around position 644 in the amino-terminal region of the CP. In this sequence, the program was unable to identify sharp breakpoints as it was overridden due to multiple recombination events. Graphic representation of the event and NJ trees of both parents clearly showed that the fragment of the recombinant strain corresponding to the minor partner has a distinct position on the phylogenetic tree (Figure 7). According to the recombination analysis LMoV strain JN127341 was a double recombinant whose major parent in event 5 was also a recombinant LMoV strain from subgroup I.

In the 4th predicted recombination event, an LMoV strain from China (acc. #: AJ310203, subgroup I) was identified. The major parent was a LMoV strain described as TBV-lily strain [30] (S44147, subgroup II) and minor parent was an LMoV strain identified in this study (acc. #: MF983709, subgroup I). The first breakpoint is at position 651 located in the amino-terminal part of the CP and the second breakpoint is at position 1584 in the 3′ end non-coding region. Graphic representation of the event and NJ trees of both parents indicated that the major parent of the recombinant grouped with the subgroup II isolates, while the fragment corresponding to the minor parent grouped with the subgroup I isolates (Figure 8). This clearly demonstrated that the recombinant resulted from a recombination between the two subgroups. 

JN127341 is potentially recombinant of subgroups I and II (event 1 and 5) and also indirectly recombinant of both subgroups by having AJ310203 as a major parent in event 5 (Figure 9). The parentage can be described as
JN127341(unknown × KF553658) × (unknown × AJ310203[MF983709 × S44147]))

#### 2.2.2. Recombination in TBV Strains

Three possible recombinants were predicted in the case of TBV; one in a TBV strain (currently accepted as *Tulip mosaic virus*) from Japan (acc. #: X63630) and two from the Netherlands (acc. #: KF440423 and KT923168).

In the case of the TBV isolate derived from Japan (2nd recombination event) (acc. #: X63630) the major parent is a Hungarian TBV isolate (MK368783) and potential minor parent is a recombinant strain from the Netherlands (KF442403). The first breakpoint is undetermined, probably at position 252, but in this sequence only the CP region was available, ending breakpoint is at position 1321 in the CP coding region. Interestingly the minor parent (acc. # KF442403) is marked as recombinant in event 3, this means X63630 is recombinant of a recombinant sequence. The graphic representation of the event and NJ trees of both parents clearly demonstrates that the fragment derived from the major and from the minor partner has distinct location (Figure 10), corresponding to the origin of the genome segments. 

The third predicted recombination event was in the 3′ UTR region of a TBV isolate derived from the Netherlands (acc. #: KF442403) In this case the major parent was a TBV strain from the present study (acc. # MK368787) and the potential minor parent is unknown, the program inferred MF983710 as minor parent. The breakpoint is at position 1598. Graphic representation of the event and NJ trees of both parents demonstrate the distinct origin of the fragments (Figure 11).

In the case of the other recombinant strain derived from the Netherlands (sixth predicted recombination event, acc. #: KT923168) the major parent is unknown, the program inferred MK368783 as major parent, potential minor parent is a TBV strain from Hungary (MK368785). The breakpoint is at position 1004. Graphic representation of the event and NJ trees of both parents and the fragments of the recombinant strain clearly demonstrates the distinct origin. The amino-terminal region of the recombinant grouped together with the major parent and the carboxy-terminal region grouped with the minor parent (Figure 12). 

### 2.3. Phylogenetic Analysis of Tulip Infecting Potyviruses Based on the Coat Protein Gene 

Phylogenetic analysis of the 32 CP sequences determined in this study and 18 CP sequences available in the GenBank database were carried out. For CP sequences the Tamura–Nei nucleotide substitution model [31] with Gamma distribution and invariant sites gave the lowest corrected Akaike Information Criterion (AICc) value, 11042.30. The phylogenetic tree was built with this model, the accuracy of the tree was tested with Bootstrap method with 1000 replicates (Figure 13). 

According to the phylogenetic tree the three viruses, TBV, LMoV, and ReTBV have clearly monophyletic origin, but are distinct species. TBV is located in a distinct clade while LMoV and ReTBV strains formed a monophyletic group supported by high bootstrap values (100%). In this monophyletic group the LMoV and the ReTBV strains formed their own monophyletic group. Both the LMoV and ReTBV isolates formed two distinct subgroups according to the high bootstrap values (100%). The two subgroups (“Tulip breaking virus lily strain” and “Tulip band breaking virus”) of LMoV were described previously by Zheng et al. [32]. However these subgroup designations are referring to obsolete species names, we provide evidence to support the findings of Rivas et al. [33] who renamed the Tulip band breaking virus group to subgroup I and Tulip breaking virus lily strain group to subgroup II of Lily mottle virus. Interestingly, all of the LMoV isolates identified in this study from tulips grouped into the LMoV subgroup I. Three isolates in this group derived from different lily species from Japan, China, and South Korea and one from tulip from Japan. Isolate acc. # AB078007 from tulip shows closest relation with other strains derived from tulip. Interestingly, in GenBank, this isolate is listed as Tulip band breaking virus, our phylogenetic analysis confirms it is indeed a strain of LMoV.

All of the isolates of subgroup II originated from different lily species and this subgroup has a worldwide distribution since it was identified in different locations, such as Australia, New Zealand, Japan, China, India, USA, and the Netherlands. One of these isolated in the Netherlands from *Lilium longiflorum* cv. Flevo (acc. #: S44147) was described as TBV Lily strain [30], but according to the close relationship with LMoV subgroup II isolates this strain should also be identified as LMoV. In the case of ReTBV, the isolates previously described from lily form a distinct clade, subgroup II. One of them was identified as Lily virus A in Australia. The other clade (subgroup I) contains strains derived from tulips.

The TBV isolates form a much more compact clade and contains all of the isolates derived from tulip from the Netherlands, Japan and Hungary.

The phylogenetic networks based on the complete nt sequences of the clones including the carboxy-terminal region of the NIb, the CP coding region and the 3′ end non-coding region (Appendix A) also supports these results. 

### 2.4. Pairwise Relationships, NIb/CP Polyprotein Cleavage Site, and Species According to the ICTV Taxonomy Criteria

Nucleotide (nt) sequence similarities of the complete CP regions of isolates used in the Maximum Likelihood (ML) phylogenetic tree were analyzed in detail. Nucleotide identities between the LMoV strains were at least 85.89%, in the case of TBV strains 94.78% while in the ReTBV isolates at least 89.38% identity have been found (Table 3). Among the different viruses the highest identity were observed between LMoV and ReTBV strains, 70.16–71.04% and TBV/ReTBV 65.95–67.13% while in the case of LMoV/TBV only 63.05–65.29% identity were detected.

Moreover, amino acid (aa) sequence similarities of the complete CP regions were analyzed. In the case among the LMoV strains, at least 87.23% identity, in the case of TBV strains, 97.01%, while in the ReTBV isolates, at least 95.49% identity have been found (Table 4). Among the different viruses the identity observed between LMoV and ReTBV strains were 66.54–72.66%, TBV/ReTBV 66.79–68.44%, and in the case of LMoV/TBV, 62.17–67.04% identity were detected.

The NIb/CP polyprotein cleavage sites were also identified, where possible. In the case of LMoV the cleavage site was VAFQ/A in all of the 21 isolates. In all of the 19 TBV isolates, VQFQ/A was the cleavage site. Among ReTBV isolates, five sequences showed VIFQ/A, while in one case VILQ/A was identified as cleavage site (Appendix A). According to the ICTV species demarcation criteria in the case of potyviruses different species should have CP aa sequence identities less than 82% and nt sequence identity less than 76% [29], and within the species the polyprotein cleavage sites should be semi-conserved. The results show that tulip infecting potyviruses clearly represent three species: TBV, LMoV, and ReTBV, in accordance with the ICTV criteria. 

## 3. Discussion

In this study, we identified and characterized tulip infecting potyviruses based on samples collected in Hungary. All of the collected tulip plants with characteristic tulip breaking symptoms were proved to be infected with potyviruses, three distinct species were identified: LMoV, TBV, and ReTBV. Evidence was provided that MAb PTY1 antibody is able to detect all of these tulip infecting potyviruses. 

According to the symptoms of infected cultivars and the causative agents, we could not link a specific symptom type to a virus or vice versa. The ReTBV was identified exclusively in the Rembrandt group of tulips. In the case of TBV and LMoV, we could not confirm host preference towards a cultivar group or cultivar. This finding contradicts current knowledge as Darwin-hybrids are thought to be not susceptible to TBV [7,13]. No double infection was detected even if both viruses were present in a single tulip population in gardens (Szada, Gödöllő). This finding is in agreement with the previous report that potyvirus populations predominantly remains separated even in infected plants [34], and in this way, spatial separation reduce opportunities for competition between viral genetic variants [35]. When looking at the virus distribution map (Figure 1), we could not substantiate any evidence of geographic distribution pattern, both LMoV and TBV was equally frequent in the different regions of Hungary.

Recombination and mutations are the two major factor in virus genome variability that is the driving force of virus evolution and variability of the viruses [36,37]. The recombination frequency depends on the degree of sequence identity between the viruses involved in the event, the length of viral genome, and the presence of recombination hot-spots [38,39,40]. In the case of potyviruses, RNA recombination constitutes an important factor promoting adaptation to changing environmental conditions and new hosts [41,42,43,44]. Interspecific recombination was detected only in a few cases, such as in the case of *Watermelon mosaic virus* [45] and *Narcissus* viruses [46], but intraspecific recombination is remarkably frequent. Recombination was also detected in the case of potyviruses: *Bean yellow mosaic virus* [47], *Canna yellow streak virus* [48], *Lettuce mosaic virus* [49], *Pea seed-borne mosaic virus* [50], *Plum pox virus* [38,51,52,53,54,55], *Potato virus Y* [56,57], *Turnip mosaic virus* [58], *Watermelon mosaic virus* [45], *Zucchini tigré mosaic virus* [59], *Zucchini yellow mosaic virus* [60,61]. Recombination hot-spots were identified in the P1 gene, but recombination in the NIb and CP regions were also identified in several cases. For example, in the case of *Plum pox virus*-TAT (PPV-TAT) isolate, recombination breakpoints have been identified at two position in the entire CP gene encompassing the region coding for hypervariable N-terminal domain [51]. Recombination in the NIb and CP C-terminal region was identified also in the case of different potato virus Y strains [56]. In the case of tulip infecting potyviruses intraspecies recombination was identified in the NIb gene, in the amino-, and also the carboxy-terminal region of the CP and in the 3′end non-coding region in this study. RDP v4.98 predicted six distinct recombination events in LMoV and TBV with at least three different methods (Table 2), each event was predicted once, and all events showed intraspecific recombination (Figure 5). In the cases of acc. # X63630 (TBV) and JN127341 (LMoV) appears to be recombination of already recombinant sequence KF442403 (TBV) and AJ310203 (LMoV). In JN127341, the program predicted two recombination events (Figure 6 and Figure 7). In the case of AJ310203 (subgroup I), the major parent (S44147) originated from subgroup II, the minor parent (MF983709) originated from subgroup I, becoming recombinant of both subgroups. This means JN127341 is directly recombinant of subgroups I and II and also indirectly recombinant of both subgroups by having AJ310203 as major parent in recombination event 5. Previously recombination suggested to be playing a role in some potyvirus evolution towards host specialization [44,47], but recently recombination was proposed as a key factor associated with expanding host range in the adaptation to new hosts [42]. 

Our results extend the reported recombinant events within the potyvirus group describing for the first time intraspecies recombination in the case of tulip infecting potyviruses.

Based on the phylogenetic tree, aa and nt identity and differences in polyprotein cleavage sites, definitely three potyvirus species were identified on tulip plants in accordance with the ICTV species demarcation criteria. The latest species demarcation criteria in the case of potyviruses require nt identity less than 76% for the complete CP region to qualify as a different species [29]. The exemplar isolate of TBV (acc. #: KF826466, RefSeq: NC_043168) contains partial cytoplasmic inclusion (CI), complete 6K2, complete viral protein genome-linked (VPg), complete nuclear inclusion A protease (NIa-Pro) and the N terminal part of the nuclear inclusion B (NIb), this isolate does not contain the CP. However, acc. # KF442403—also from 1986 June, collected by Lemmers from cv. Texas Flame—does contain the CP. The type isolate of *Tulip mosaic virus* (acc. #: X63630, RefSeq: NC_043425) and TBV both are present as separate species in the ICTV database, GenBank, and Master Species List 2019.v1, but the two isolates (acc. # X63630 and KF442403) have high nt sequence identity (98.78%). Moreover, X63630 has 96.96% nt identity with TBV isolate BC24 (acc. #: MH886517) (Alignment, Appendix A), so *Tulip mosaic virus* should be degraded to synonym of TBV. The formerly described Tulip band breaking virus (acc. #: AB090385) and Tulip breaking virus-lily strain (acc. #: S44147) branch with LMoV strains, which is in agreement with previous reports [19,20,21,22], so they are clearly strains of LMoV. The location on the phylogenetic tree also confirm the identity of these isolates. 

Based on the nt and aa sequence identities and phylogenetic analysis presented in this study, ReTBV should be elevated to species level (assigned as distinct species). Lily virus A (acc. #: JN127335) has 89.38 to 99.88% nt, and at least 95.49% aa identity in the CP region, with the ReTBV isolates described in this study (acc. #: MK368780, MK368781, MK368782), and also with the previously identified ReTBV from tulip derived from the Netherlands (acc. #: MG637051) and from *Lilium* Asiatic hybrid analyzed in Japan (acc. #: AB674535), so Lily virus A should be a synonym of ReTBV.

The purpose of a phylogeny is to describe the evolutionary history of biological entities. According to the phylogenetic analysis of the tulip infecting potyviruses, the TBV isolates clearly form a distinct clade while LMoV and ReTBV strains formed a monophyletic group supported by high bootstrap values (100%). Within this monophyletic group the ReTBV and LMoV strains formed their own monophyletic group and they share a common ancestor. Our results support that LMoV isolates can be divided into two subgroups. Interestingly all of the strains isolated from tulip in Hungary and Japan grouped to subgroup I and in this subgroup only three isolates originated from *Lilium* species. In subgroup II all of the isolates originated from *Lilium* species. The isolates of ReTBV identified from tulips and lilies each also form a separate clade similarly to the LMoV strains. Due to the high bootstrap value, we propose to emerge subgroup I and II in ReTBV. Since limited data is available. it is in question whether real biological separation is in progress, so in the future, characterization of increasing number of isolates may answer this question. Previous analysis of a worldwide collection of *Turnip mosaic virus* (genus *Potyvirus*) demonstrated four phylogenetic lineages, mostly congruent with host types [62,63], so the observed separations could be a result of adaptation to the host species. 

The extended analysis of potyviruses infecting monocotyledonous plants is primary interest, since monocotyledonous bulbs and grasses from Western Eurasia were assumed to be the primary and ancestral host for all potyviruses [41,64,65]. Further survey of viruses on monocotyledonous hosts might reveal alteration of potyvirus populations and provide insight into the ancient divergence of potyviruses as part of their evolution.

## 4. Materials and Methods 

### 4.1. Plant Samples

Samples were collected during the flowering period of tulip plants showing tulip breaking syndrome from home and public gardens in different regions of Hungary during the growing seasons of 2017 and 2018. Permission to collect samples were asked in every case from the owners or head gardeners. For spatial distribution of viruses the samples were collected primarily from cultivar Apeldoorn (Darwin-hybrid group) [5,66], the most widely planted cultivar in Hungary. Several samples from different cultivars were also collected from a private tulip collection in Szada to recognize the occurrence of different virus species in a tulip population. Sample details, the location and the date of the collection and the tulip cultivars are presented in Appendix A. 

One gram of symptomatic leaf tissue was collected from each variety. Collected samples were transported on ice. Storage was carried out in a deep freezer at −70 °C ± 2 °C. 

### 4.2. Enzyme-Linked Immunosorbent Assay (ELISA)

Potyvirus specific ELISA were carried out from the samples: antigen coated plate ELISA based on monoclonal antibody MAb PTY1 (Agdia Cat # 27200, RRID:AB_2819158) [67] were carried out on each sample in duplicates according to the manufacturer’s instructions. Leaf tissue was homogenized in extraction buffer in 1:100 (m:v) dilution. The monoclonal detection antibody (Batch # 02491, 02495) was raised in mouse against potyvirus coat protein (Clone PTY1), the enzyme conjugate (Batch # 02368, 02372) was a polyclonal antibody to mouse IgG and was raised in rabbit. Positive control was provided by the manufacturer (Cat # 20001, Lot # C2130), negative control was prepared from *Chenopodium amaranticolor* seedlings grown in a vector free greenhouse. The final dilution of both the detection antibody and the enzyme conjugate were 1:100 (*v*:*v*). Plates were read at 405 nm wavelength on a Labsystems Multiskan MS ELISA reader after 15, 30, and 60 min incubation. A sample was considered positive if the absorbance of the sample was at least three times greater than that of the negative control.

### 4.3. RNA Extraction, Reverse Transcription, PCR Cloning and Nucleotide Sequence Determination

Total RNA extraction was carried out according to the protocol of White and Kaper [68]. Total RNA was stored at −70 °C ± 2 °C until further processing. First strand cDNA was synthesized according to Salamon and Palkovics [69] with universal potyvirus reverse primer poly T_2_ (5′-CGGGGATCCTCGAGAAGCTTTTTTTTTTTTTTTTT-3′). PCR reaction for 50 µL contained the following ingredients: 2 µL cDNA, 5 μL 10× Taq Buffer with KCl (Thermo Fisher Scientific Baltics UAB, Vilnius, Lithuania Cat # EP0402), 3 μL 25 mM MgCl_2_, 2 μL dNTPs, 10 mM each, 1-1 μL 100 µM poty7941 forward primer (5′-GGAATTCCCGCGGNAAYAAYAGYGGNCARCC-3′) and poly T_2_ as described earlier [69], 1 μL *Taq* DNA Polymerase (recombinant) (Thermo Fisher Scientific Baltics UAB, Cat # EP0402), and 35 μL nuclease free H_2_O. The PCR program for potyvirus detection started with initial denaturation for 3 min at 94 °C. The cycle continued with denaturation for 30 s at 94 °C, annealing primers for 30 s at 50 °C, polymerization at 72 °C for 2 min. The cycle was repeated 40 times, and a final polymerization step was included, 10 min at 72 °C. PCR products were visualized on 1% TBE agarose gels. *Potato virus Y* (acc. #: M95491) [70] maintained in bell peppers was used as positive control. The primers amplified the C terminal part of the RNA-dependent RNA polymerase (NIb) including the highly conserved GNNSGQP motif, the complete coat protein (CP) sequence, the complete 3′ untranslated region (UTR) to the first few bases of the polyA tail. PCR products ranging from 1650 to 1700 base pairs (bp) were purified with High Pure PCR Product Purification Kit (Roche, Mannheim, Germany), according to the manufacturer’s instructions. The PCR products were ligated into pGEM^®^-T Easy vector (Promega, Madison, WI, USA, Cat # A1360) according to the manufacturer’s instruction. Plasmids were recovered with GeneJET Plasmid Miniprep Kit (Thermo Fisher Scientific Baltics UAB, Cat # K0502) and the sequence determination of the clones were carried out by service providers Biomi (Gödöllő, Hungary) and Base Clear BV. (Leiden, the Netherlands). To analyze whether a single potyvirus is present in the tulip plants, all of the original PCR products were digested with specific restriction endonucleases: in the case of TBV: *Nhe*I, ReTBV: *Hind*III, LMoV: *Sac*II.

### 4.4. Sequence Analysis and Comparisons

Each read was compared to sequences in the National Center for Biotechnology Information, U.S. National Library of Medicine (NCBI) GenBank using megaBLAST module [71] to identify it to species level [29,72]. Sequence reads were mapped to a reference using Unipro UGENE v1.30 [73]. Trimming quality threshold was set to 0, mapping minimum similarity to 50%. For ReTBV the reference was acc. #: JN127335, for TBV KF442403 and AB054886 for LMoV. From each sequence the adapter regions of the primers (5′-GGAATTCCCG-3′ for poty7941 and 5′-CGGGGATCCTCGAGAAGC-3′ for poly T_2_) were cut off, assembled sequences were re-checked with megaBLAST to confirm identity.

### 4.5. Phylogenetic Analysis

For phylogenetic analysis alignment was built from the nucleotide (nt) sequences of acc. #: AB053256, AB078007, AB090385, AB674535, AF531458, AJ310203, EU267778, FJ618539, JN127335, JN127341, KF442403, KF553658, KJ561805, KT923168, MG637051, MF781080, MH360239, NC_001555, S44147, and X63630 using MEGA X [74]. ClustalW [75] alignments were built from the nt sequences of the CP regions of the previously listed accession numbers, and the ones we determined in this study. Default settings were used in the program, except DNA weight matrix was set to ClustalW (1.6). After aligning the sequences, we selected the best fit DNA model. We used the model with the lowest corrected Akaike Information Criterion (AICc) value [76] to construct Maximum Likelihood (ML) tree [77,78] for CP nt sequences. The reliability of the tree was tested with the Bootstrap method [79] with 1000 replications. For better understanding of the possible evolutionary scenarios we used SplitsTree v4.15.1 [80] to build a phylogenetic network from the alignment of the regions our primers amplified with default settings. In all of the experiments, *Tobacco etch virus* (RefSeq: NC_001555) was used as outgroup.

Sequence identities were calculated in the alignment of CP sequences in MEGA X, using the p-distance matrix output.
Identity% = (1 − (p-distance)) × 100.

In the case of *Tulip mosaic virus* and *Tulip breaking virus* we used the following GenBank accession numbers for nucleotide alignment: KF442402, KF442403, KF826466, KT923168, MF983710, MH886517, MK368783, MK368785, MK368786, MK368787, MK368789, MK368791, MK368795, MK368796, MK368797, MK368799, MK368800, MK368805, MK368807, MK368808, X63630. The alignment was generated in CLC Genomics Workbench v20.0 (Qiagen) with less accurate option.

For further evidence of taxonomic relations, the amino acid sequences at the NIb/CP cleavage sites were compared, where available. According to Adams et al. [81], we provide the summary of amino acid sequences from P4 to P1′ positions of sequences used to build the ML tree.

### 4.6. Recombination Detection

For recombination prediction the RDP v4.98 program [82,83] was used on the alignment generated for SplitsTree with the following settings: sequences were set to linear, masking was automatic. Automatic exploratory analysis was carried out with RDP [84], Chimaera [85], Bootscan/Recscan [86], 3seq [87], GENECONV [88], MaxChi [89], and SiScan [90] methods. For Bootscan/Recscan method Jin and Nei [91] substitution model was used. MaxChi and Chimaera were set to variable window size, fraction of variable sites per window was 0.1, SiScan *p*-value permutation number was 447, PhylPro [92] was instructed not to use self-comparison. LARD [93] was run with Reversible process [94] and moving partition scan. TOPAL [95] with Jin and Nei model and transition:transversion ratio 0.5. Distance plots and Neighbor Joining (NJ) trees [96] also used the Jin and Nei model, with 100 Bootstrap replicates [79], random number seed was 3. SCHEMA [97] used interaction distance 4, temperature 20, sequences were set to RNA. The rest of the options were left default. After the exploratory run, all predicted recombination events were re-checked with all methods. Then each event was checked manually if the trees were topologically different, and for any variation on predicted breakpoints, and corrected if necessary. Events detected with less than three methods were rejected. 

## Figures and Tables

**Figure 1 plants-09-01807-f001:**
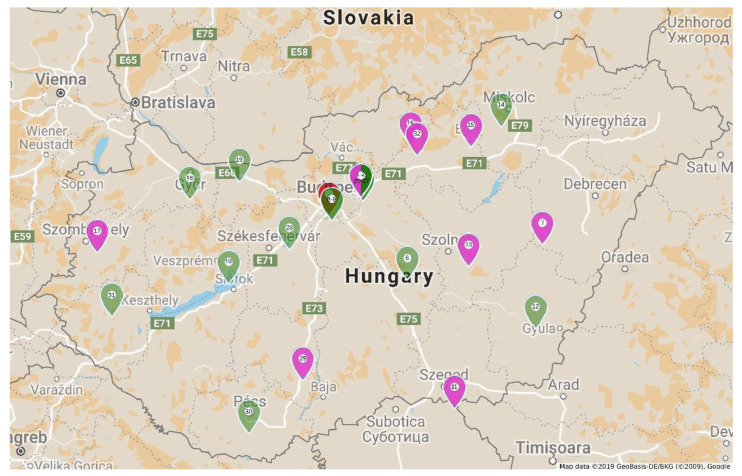
Location of collected samples showing tulip breaking syndrome in Hungary. Sample numbers are placed on markers. Red marker indicates Rembrandt tulip-breaking virus (ReTBV), green *Tulip breaking virus* (TBV), pink *Lily mottle virus* (LMoV) infection. Bicolor marker represents tulip population with TBV and LMoV infection.

**Figure 2 plants-09-01807-f002:**
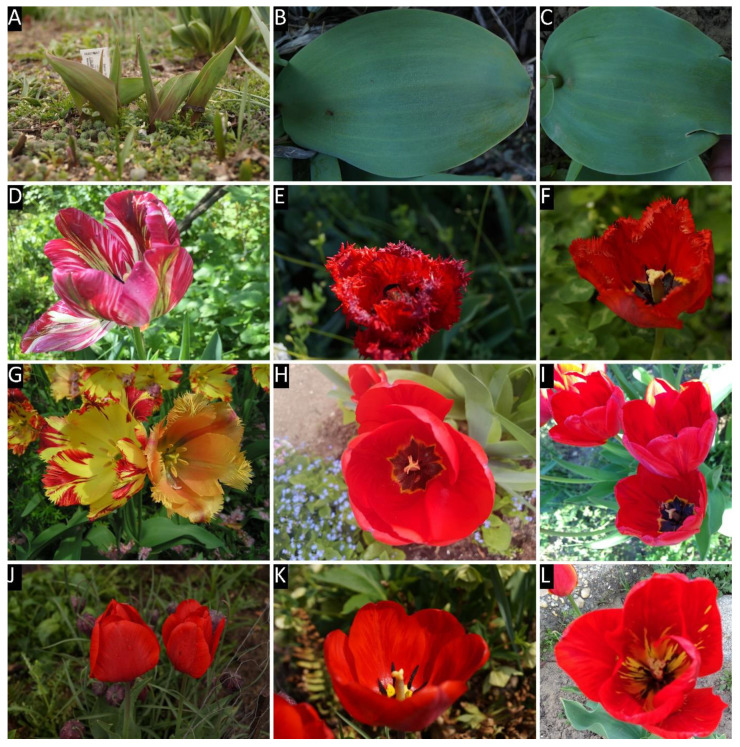
Symptoms of tulip breaking virus on different cultivars. Panel labels are followed by cultivar names; isolate numbers and acc. # are in brackets. (**A**) Barbados (675, MK368783), (**B**) Apeldoorn (713, MK368789), (**C**) Apeldoorn (738, MK368808), (**D**) Groenland (A1, MF983710), (**E**) Barbados (675, MK368783), (**F**) Crystal Beauty (705, MK368786), (**G**) Lambada, left infected, right healthy flower (704, MK368785), (**H**) Apeldoorn (707, MK368787), (**I**) Apeldoorn (722, MK368797), (**J**) Apeldoorn (721, MK368796), (**K**) Apeldoorn (713, MK368789), (**L**) Oscar (723, MK368798).

**Figure 3 plants-09-01807-f003:**
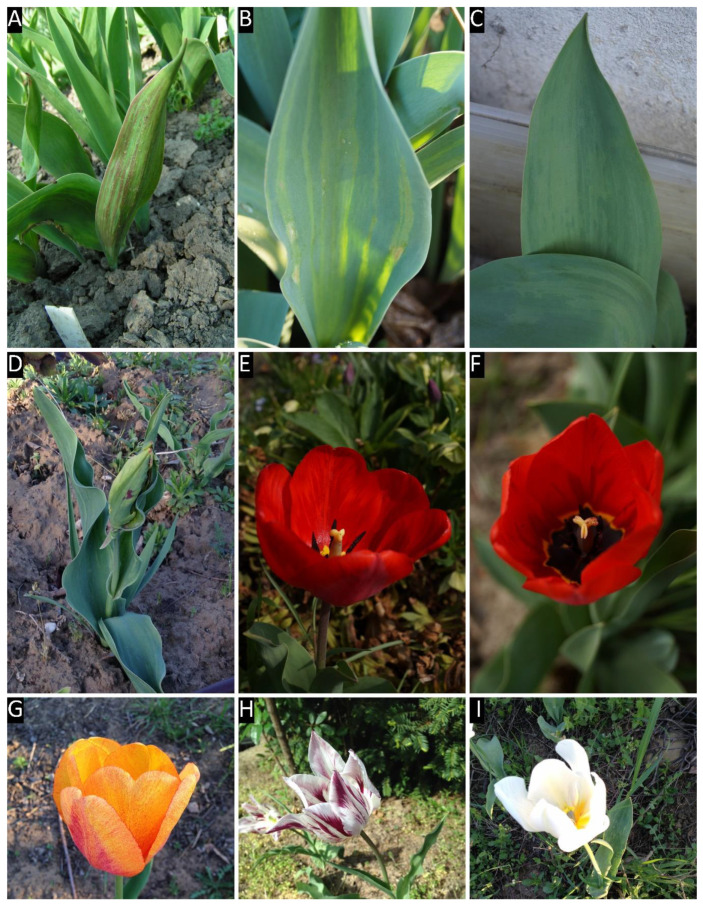
Symptoms of lily mottle virus on different cultivars. Panel labels are followed by cultivar names; isolate numbers and acc. # are in brackets. (**A**) Apeldoorn (696, MK368784), (**B**) Apeldoorn (717, MK368792), (**C**) Apeldoorn (714, MK368790), (**D**) Texas Gold (732, MK368804), (**E**) Apeldoorn (696, MK368784), (**F**) Apeldoorn (712, MK368788), (**G**) Gudoshnik (728, MK368802), (**H**) Claudia (739, MK368809), (**I**) Purissima (731, MK368803).

**Figure 4 plants-09-01807-f004:**
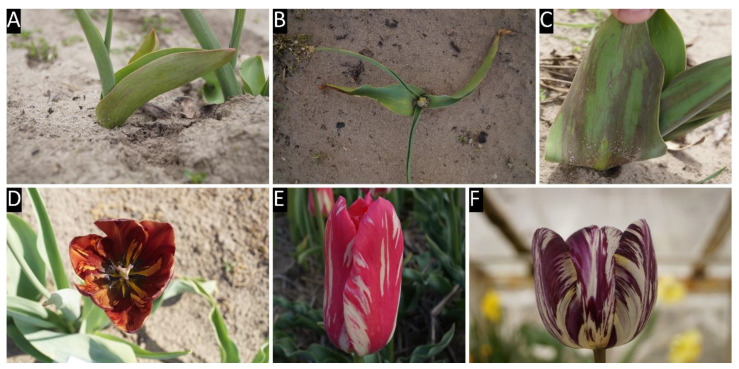
Symptoms of Rembrandt tulip-breaking virus on different cultivars. Panel labels are followed by cultivar names; isolate numbers and acc. # are in brackets. (**A**) Absalon (642, MK368780), (**B**) Zomerschoon (645, MK368781), (**C**) Insulinde (646, MK368782), (**D**) Absalon (642, MK368780), (**E**) Insulinde (646, MK368782), (**F**) Zomerschoon (645, MK368781).

**Figure 5 plants-09-01807-f005:**
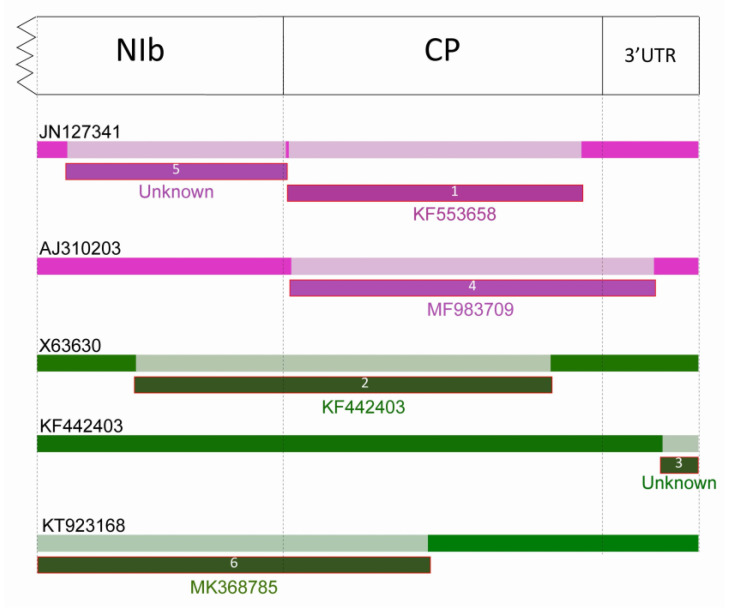
Schematic display of predicted recombination events on the analyzed region of potyvirus genome map. First dashed line indicates the beginning of the cloned fragment (GNNSGQP motif). Second dashed line indicates the NIb/CP cleavage site, the third line the stop codon of the polyprotein and the last represents the end of the 3’ untranslated region (3′ UTR) preceding the polyA tail. Arabic numbers in the bars indicate the order in which RDP characterized the event. Accession numbers above the bars (black numbers) indicate the recombinant sequence. Accession numbers and “Unknown” under the number of the recombination event represents the minor parents. Schematic display of sequences are generated by the program, bars have been consistently colored according to the virus species. Pink-purple bars (recombination events 1, 4, and 5) indicate LMoV, green bars (recombination event 2, 3, and 6) indicate TBV sequences. The breakpoints are always mentioned according to the position in the alignments.

**Figure 6 plants-09-01807-f006:**
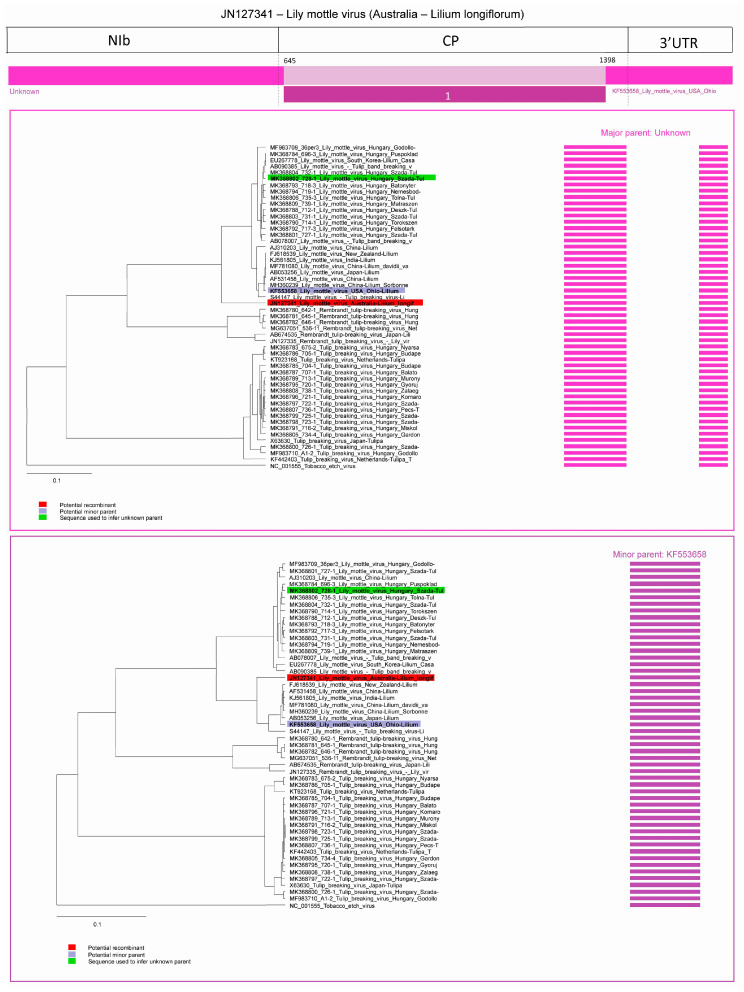
Schematic display of recombination event 1. Neighbour Joining (NJ) tree of inferred major parent is on the left, minor parent is on the right. Bars to the right of the trees represent the sequence regions from which the trees were built. All bars have been recolored to match major and minor parent colors for clarity. Red background indicates recombinant sequence, the lilac background indicates minor parent and light green inferred major parent.

**Figure 7 plants-09-01807-f007:**
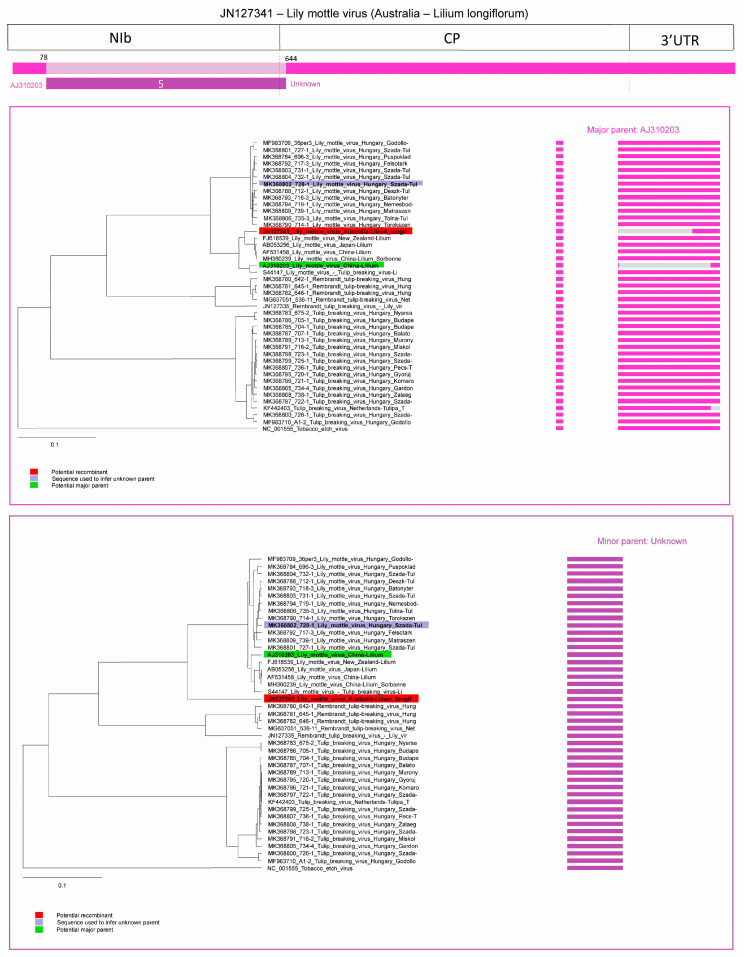
Schematic display of recombination event 5. NJ tree of inferred major parent is on the left, minor parent is on the right. Bars to the right of the trees represent the sequence regions from which the trees were built. All bars have been recolored to match major and minor parent colors for clarity. Red background indicates recombinant sequence, lilac inferred minor parent and light green major parent.

**Figure 8 plants-09-01807-f008:**
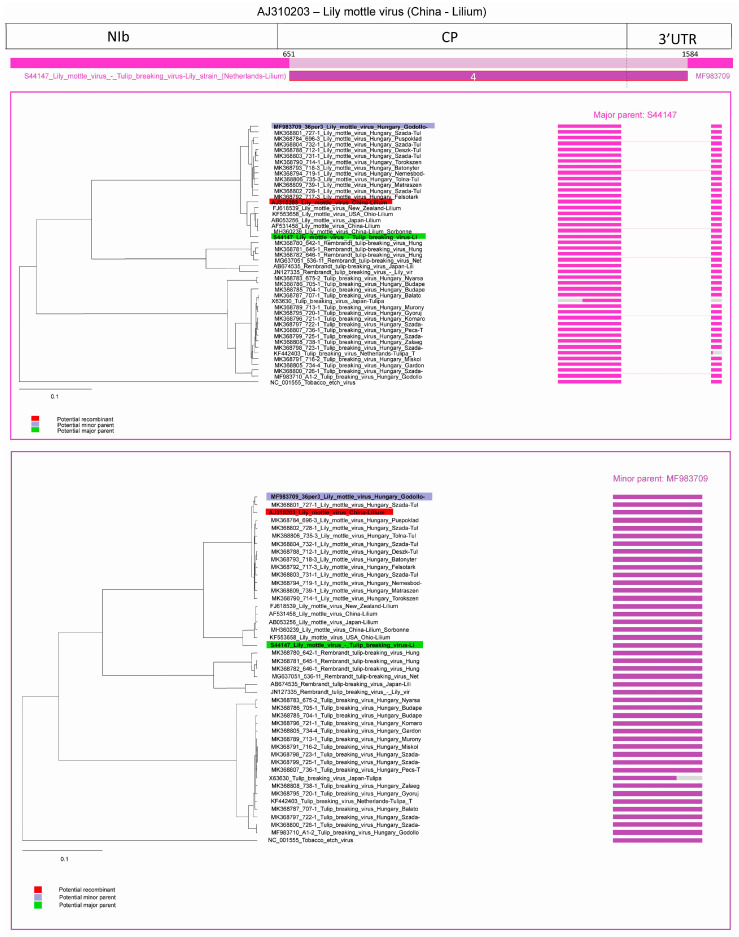
Schematic display of recombination event four. NJ tree of inferred major parent is on the left, minor parent is on the right. Bars to the right of the trees represent the sequence regions from which the trees were built. All bars have been recolored to match major and minor parent colors for clarity. Red background indicates recombinant sequence, lilac indicates minor parent and light green major parent.

**Figure 9 plants-09-01807-f009:**
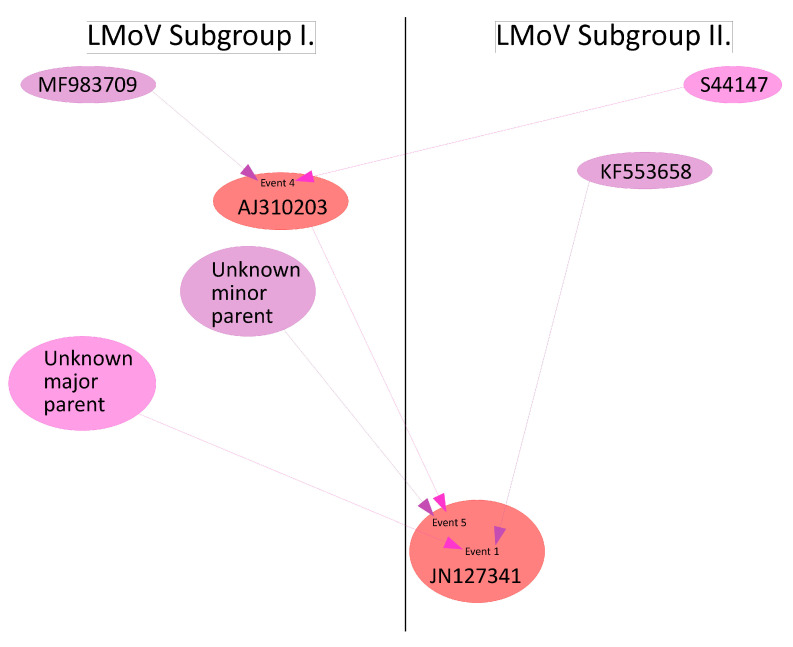
Schematic representation of the potential lineage of JN127341. Pink arrows and ellipses represent major parents, purple arrows, and ellipses represent minor parents. Red ellipses denote recombinant sequences.

**Figure 10 plants-09-01807-f010:**
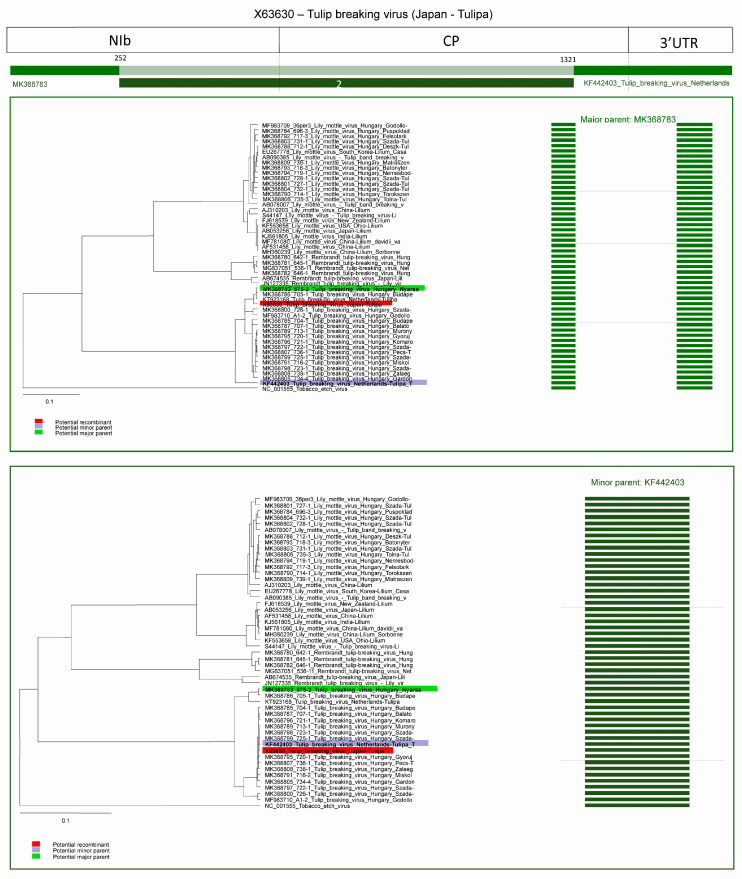
Schematic display of recombination event 2. NJ tree of inferred major parent is on the left, minor parent is on the right. Bars to the right of the trees represent the sequence regions from which the trees were built. All bars have been recolored to match major and minor parent colors for clarity. Red background indicates recombinant sequence, lilac minor parent, and light green major parent.

**Figure 11 plants-09-01807-f011:**
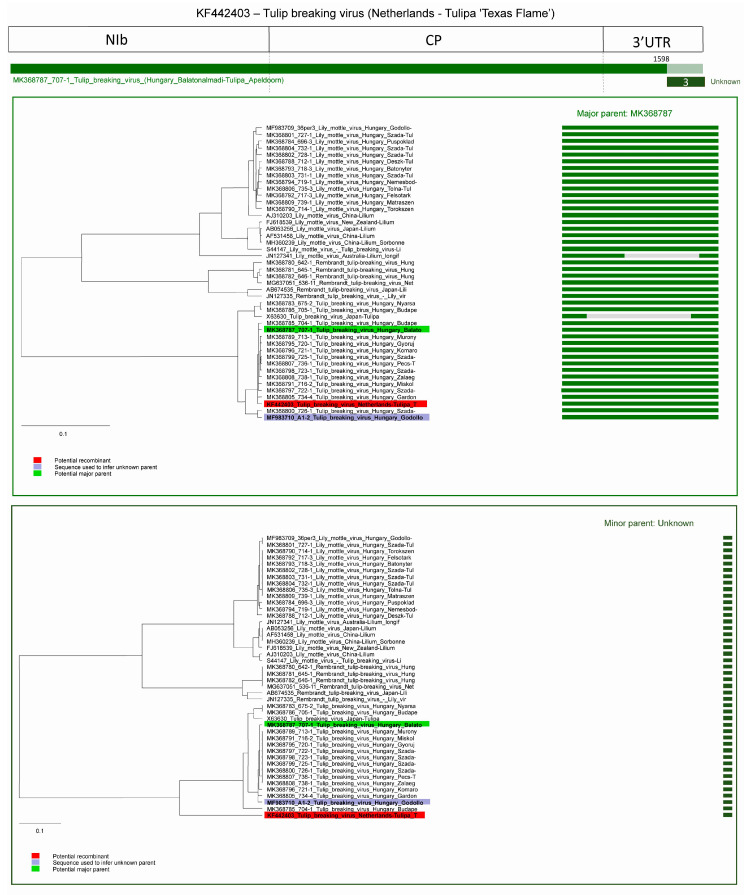
Schematic display of recombination event 3. NJ tree of inferred major parent is on the left, minor parent is on the right. Bars to the right of the trees represent the sequence regions from which the trees were built. All bars have been recolored to match major and minor parent colors for clarity. Red background indicates recombinant sequence, lilac inferred minor parent, and light green major parent.

**Figure 12 plants-09-01807-f012:**
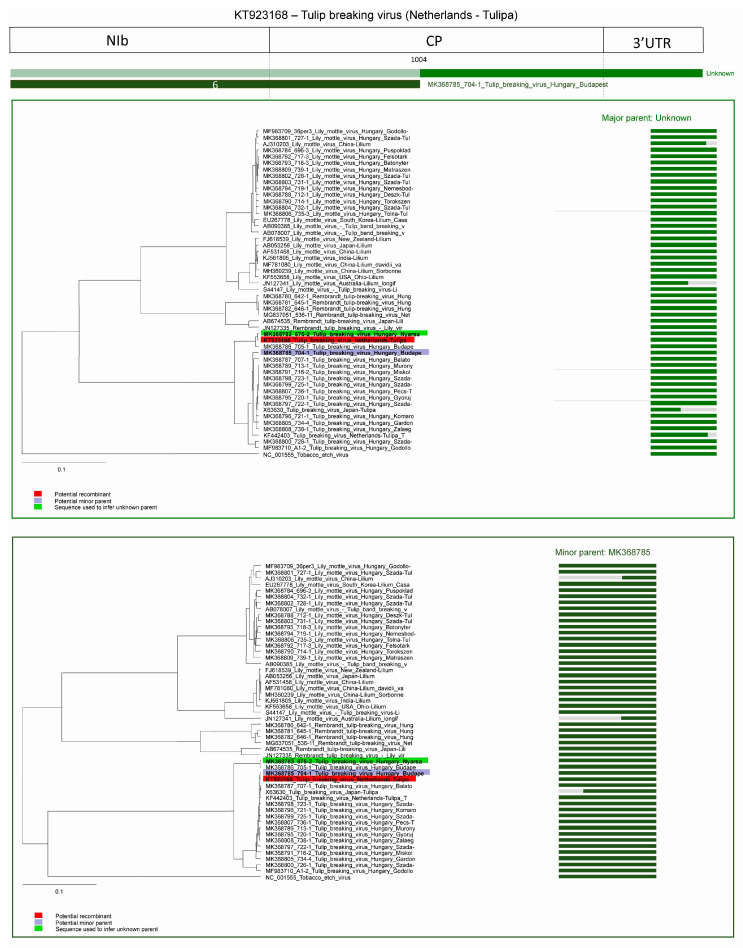
Schematic display of recombination event 6. NJ tree of inferred major parent is on the left, minor parent is on the right. Bars to the right of the trees represent the sequence regions from which the trees were built. All bars have been recolored to match major and minor parent colors for clarity. Red background indicates recombinant sequence, lilac minor parent, and light green inferred major parent.

**Figure 13 plants-09-01807-f013:**
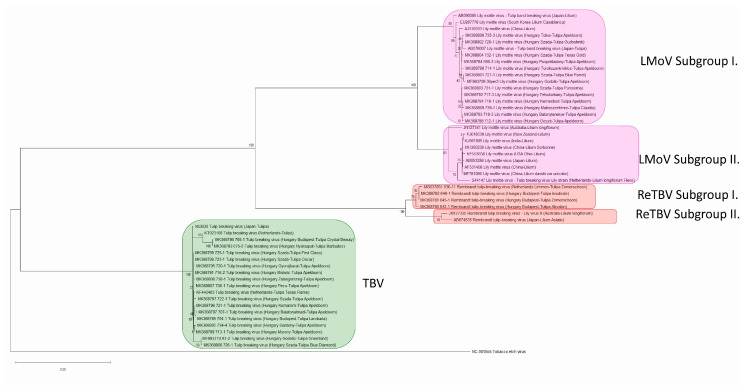
Maximum Likelihood tree of CP sequences. Built with Tamura–Nei nucleotide substitution model with Gamma distribution, invariant sites and tested with the Bootstrap method with 1000 replicates. Outgroup is *Tobacco etch virus* (RefSeq: NC_001555). Bootstrap values are indicated on the branches as percent.

**Table 1 plants-09-01807-t001:** Summary of predicted recombination events.

Recombination Event	Acronym of Recombinant Virus	Recombinant Sequence	Major Parent	Minor Parent
1	LMoV	**JN127341**	unknown	KF553658
2	TBV	X63630	MK368783 ^1^	KF442403 ^2^
3	TBV	KF442403	MK368787 ^1^	unknown
4	LMoV	AJ310203	S44147	MF983709 ^1^
5	LMoV	**JN127341**	AJ310203 ^2^	unknown
6	TBV	KT923168	unknown	MK368785 ^1^

^1^ indicates sequences described in this article. ^2^ indicates recombinant parent. Underlining indicates recombinant sequence with recombinant parent. Bold type indicates more than one recombination event is present in the sequence.

**Table 2 plants-09-01807-t002:** Summary of confirmation tables for recombination events 1 to 6.

	Average *p*-Values ^1^ by Recombination Event Number					
	Recombination Event					
Method	1	2	3	4	5	6
RDP	1.713 × 10^−5^	5.965 × 10^−9^	1.655 × 10^−6^	1.863 × 10^−1^	5.776 × 10^−1^	2.448 × 10^−2^
GENECONV	7.062 × 10^−9^	4.312 × 10^−7^	3.589 × 10^−5^	3.589 × 10^−2^	-	4.265 × 10^−1^
Bootscan	3.197 × 10^−5^	>1.0	>1.0	2.2 × 10^−3^	-	>1.0
MaxChi	>1.0	>1.0	>1.0	>1.0	>1.0	6.678 × 10^−1^
Chimaera	9.692 × 10^−10^	5.254 × 10^−9^	>1.0	>1.0	1.087 × 10^−3^	6.317 × 10^−1^
SiScan	6.921 × 10^−16^	2.561 × 10^−26^	7.68 × 10^−17^	4.867 × 10^−27^	5.221 × 10^−20^	3.991 × 10^−7^
PhylPro	5.212 × 10^−20^	1.123 × 10^−19^	1.489 × 10^−13^	9.194 × 10^−8^	2.279 × 10^−7^	3.827 × 10^−5^
LARD	-	-	-	-	-	-
3Seq	-	-	-	-	-	-

^1^ Bonferroni corrected. Highest acceptable *p*-value was 0.05.

**Table 3 plants-09-01807-t003:** Summary of the nt identity of complete CP region of accession numbers used to build Maximum Likelihood (ML) tree.

	Identities of Virus nt Sequences (%)			
Acronym	LMoV	TBV	ReTBV	TEV
LMoV	85.89–99.76%			
TBV	63.05–65.29%	94.78–100%		
ReTBV	70.16–71.04%	65.95–67.13%	89.38–99.88%	
TEV	59.95–60.84%	63.91–65.82%	60.71–60.91%	100%

Data compiled from Appendix A Identity% sheet.

**Table 4 plants-09-01807-t004:** Summary of the amino acid (aa) identity of complete CP region of accession numbers used to build ML tree.

	Identities of Virus aa Sequences (%)			
Acronym	LMoV	TBV	ReTBV	TEV
LMoV	87.23–100%			
TBV	62.17–67.04%	97.01–100%		
ReTBV	66.54–72.66%	66.79–68.44%	95.49–100%	
TEV	58.56–59.32%	64.26–65.01%	61.24–62.16%	100%

Data compiled from Appendix A Identity% sheet.

## Data Availability

Data are contained within the article or supplementary material.

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
