# Peer review of "Genetic Diversity of Potyviruses Associated with Tulip Breaking Syndrome"

_plants, 2020, doi:10.3390/plants9121807_

Round 1
Reviewer 1 Report
This manuscript presents a generally thorough examination of the presence and relationships of the potyviruses infecting tulips in Hungary, and as a result propose that Rembrandt tulip breaking virus (ReTBV) should be re-established as an independent species of the genus Potyvirus. However, as the elevation of ReTBV back to full species status forms a significant portion of the work, it is peculiar that not all available ReTBV sequences including the full coat protein (CP) gene are included in the analysis; inexplicably missing from the analysis is ReTBV isolate 536-11 (MG637051), which was collected by lead author János Ágoston in the Netherlands in 2014, from Rembrandt tulip variety ‘Zomerschoon’ – the same variety originating in 1620 in which isolate 645-1 (MK368781) was collected in Hungary in 2018. Notably isolate 536-11 and 645-1 differ at least in one point which the authors thought worthy of note in regard to the other isolates; isolate 645-1 (MK368781) has the predicted CP cleavage site VILQ/A, whereas the other four isolates (including that described as ‘Lily virus A’, from lily in Australia; JN127335) have VIFQ/A – as does isolate 536-11 (MG637051). As the two isolates from ‘Zomerschoon’ were collected in different countries, and four years apart, and differ in at least this detail (though otherwise sharing over 99.2% nt identity), it is hard to understand this omission, which should probably be rectified. I would also recommend – though not for inclusion in this manuscript – that the authors obtain the full sequence – or at least, the full coding sequence – of one of the ReTBV isolates in order to provide a true reference sequence for ReTBV and cement its status as a distinct potyvirus species.
Beyond the omission of ReTBV isolate 536-11 (MG637051) from the analysis, my main issue with the manuscript is that the Figures 6, 7, 8, 10, 11, 12 and 13 are presented at a size which makes it essentially impossible to read from the printed page. The text in these figures is hard to read even when enlarged on screen so that only one panel (either left or right) fills the full screen width; the color coding used to indicate the positions of recombinant sequences and the predicted major and minor parental sequences provides the only real way to make sense of these figures. I would also suggest that the cartoon representation of the sequence fragment above each of Figures 6, 7, 8, 10, 11 and12 should be modified to show the nucleotide numbers at the start and end of the sequences, plus at the breakpoints – and also to indicate the NIb:CP and CP:3ʹ-UTR junctions. One possible means of making these figures more readily legible would be to place the panels of each figure stacked as top and bottom, instead of side by side - at the cost of using more page space – but making the figures more valuable to the reader. Alternatively these figures could be presented as Supplemental Figures (but still in the ‘stacked’ rather than ‘side-by-side’ format, so as to allow maximum image size on the page).
One sentence in the legends to each of Figures 6, 7, 8, 10, 11 and12 should be modified – to read “Bars to the right of the trees represent the sequence regions from which the trees were built.”
Figure 9 – the red ellipse at the bottom of the figure is mis-labeled; JN123341 is the rRNA sequence of an uncultured bacterium – it should be JN127341.
Figure 13 – At a minimum the font size for bootstrap values should be increased to be visible; they are unreadable even at full page width on-screen. A supplemental table indicating which LMoV and which ReTBV isolates fall into the respective subgroups (and also showing their hosts) would be useful to support the Discussion on these subclades.
Figure S3 – According to the Figure legend, the ‘missing’ ReTBV sequence is included here – but MG637051 is NOT listed in the Materials and Methods at l.500-502.
Minor language corrections:
l.42 better as just ‘the first well documented record’; the ‘very’ adds little.
l.62 ‘from tulips, including tulip breaking virus …’
l.86 ‘inconsistencies’ (plural)
l.90 ‘taxonomic relationships’
l.105 ‘were present at diverse locations’
l.205 ‘showed that the fragment’
l.206 ‘minor partner has a distinct position’
l.207 ‘was a double recombinant’
l.292 ‘TBV is located in a distinct clade’
l.325 ‘Nucleotide identities between the LMoV strains’
l.343 ‘Among ReTBV isolates four sequences showed VIFQ;A’ – and note ‘four’ isolates, not ‘three’!
l.351 ‘samples collected in Hungary.’
l.383 ‘RDP v4.98 detected six distinct’ (delete ‘have’, spell out numbers up to ten unless followed by a unit of measure)
l.392 ‘suggested to be playing a role in’
l.420-421 ‘isolates clearly form a distinct clade’
l.422 ‘Within this monophyletic group’
l.427 ‘tulips and lilies each also form a separate’
l.460 ‘amaranticolor’ (delete the ‘h’)
l.461 ‘Plates were read at’
l.462 ‘wavelength on a Labsystems Multiskan’
l.500-502 Missing ‘MG637051’ – used at least for the phylogenetic network (Figure S3)
l.553 ‘Alignment S7’ (not ‘S6’)
I would be pleased to recommend publication of a suitably revised version of this manuscript.
Author Response
Response to Editors and Reviewers’ Comments
Manuscript ID: plants-1035725
Title: Genetic diversity of potyviruses associated with tulip breaking syndrome
Dear Ms. Juicy Yang,
We would like to thanks the Associate Editor and the Reviewers for the effort and time put into the review of our manuscript (MS).
We have carefully considered the comments and we provide responses all to them.
Sincerely yours,
Laszlo Palkovics
Comments from Reviewer #1:
- „ .. it is peculiar that not all available ReTBV sequences including the full coat protein (CP) gene are included in the analysis; inexplicably missing from the analysis is ReTBV isolate 536-11 (MG637051), which was collected by lead author János Ágoston in the Netherlands in 2014, from Rembrandt tulip variety ‘Zomerschoon’ – the same variety originating in 1620 in which isolate 645-1 (MK368781) was collected in Hungary in 2018. Notably isolate 536-11 and 645-1 differ at least in one point which the authors thought worthy of note in regard to the other isolates; isolate 645-1 (MK368781) has the predicted CP cleavage site VILQ/A, whereas the other four isolates (including that described as ‘Lily virus A’, from lily in Australia; JN127335) have VIFQ/A – as does isolate 536-11 (MG637051). As the two isolates from ‘Zomerschoon’ were collected in different countries, and four years apart, and differ in at least this detail (though otherwise sharing over 99.2% nt identity), it is hard to understand this omission, which should probably be rectified…”
Answer: We have rebuilt the ML tree including 536-11 isolate of ReTBV. We also added the accession to the Materials and Methods section.
- „..I would also recommend – though not for inclusion in this manuscript – that the authors obtain the full sequence – or at least, the full coding sequence – of one of the ReTBV isolates in order to provide a true reference sequence for ReTBV and cement its status as a distinct potyvirus species.”
Answer: We would like to continue our research in this direction, but it depends on the available funds.
- „Beyond the omission of ReTBV isolate 536-11 (MG637051) from the analysis, my main issue with the manuscript is that the Figures 6, 7, 8, 10, 11, 12 and 13 are presented at a size which makes it essentially impossible to read from the printed page. The text in these figures is hard to read even when enlarged on screen so that only one panel (either left or right) fills the full screen width; the color coding used to indicate the positions of recombinant sequences and the predicted major and minor parental sequences provides the only real way to make sense of these figures. I would also suggest that the cartoon representation of the sequence fragment above each of Figures 6, 7, 8, 10, 11 and12 should be modified to show the nucleotide numbers at the start and end of the sequences, plus at the breakpoints – and also to indicate the NIb:CP and CP:3ʹ-UTR junctions. One possible means of making these figures more readily legible would be to place the panels of each figure stacked as top and bottom, instead of side by side - at the cost of using more page space – but making the figures more valuable to the reader...”
Answer: We have made new figures based on the suggestions of the reviewer.
- „One sentence in the legends to each of Figures 6, 7, 8, 10, 11 and12 should be modified – to read “Bars to the right of the trees represent the sequence regions from which the trees were built.”
Answer: Corrected.
- „Figure 9 – the red ellipse at the bottom of the figure is mis-labeled; JN123341 is the rRNA sequence of an uncultured bacterium – it should be JN127341.”
Answer: The figure and all mis-typed mentions of JN127341 have been corrected.
“Figure 13 – At a minimum the font size for bootstrap values should be increased to be visible; they are unreadable even at full page width on-screen.”
Answer: We were not able to increase the font size on the tree as MEGA X does not support this. We only could export the tree in graphical formats.
- ”A supplemental table indicating which LMoV and which ReTBV isolates fall into the respective subgroups (and also showing their hosts) would be useful to support the Discussion on these subclades.”
Answer: We prepared the requested table as an excel file: S8_Table.xlsx
- “Figure S3 – According to the Figure legend, the ‘missing’ ReTBV sequence is included here – but MG637051 is NOT listed in the Materials and Methods at l.500-502.”
Answer: We have included this accession in the Materials and Methods.
- “Minor language corrections:
42 better as just ‘the first well documented record’; the ‘very’ adds little.”
Answer: Corrected.
- “62 ‘from tulips, including tulip breaking virus …’”
Answer: Corrected.
- “86 ‘inconsistencies’ (plural)”
Answer: Corrected.
- “90 ‘taxonomic relationships’”
Answer: Corrected.
- “105 ‘were present at diverse locations’”
Answer: Corrected.
- “205 ‘showed that the fragment’”
Answer: Corrected.
- “206 ‘minor partner has a distinct position’”
Answer: Corrected.
- “207 ‘was a double recombinant’”
Answer: Corrected.
- “292 ‘TBV is located in a distinct clade’”
Answer: Corrected.
- “325 ‘Nucleotide identities between the LMoV strains’”
Answer: Corrected.
- “343 ‘Among ReTBV isolates four sequences showed VIFQ;A’ – and note ‘four’ isolates, not ‘three’!”
Answer: Corrected to five, as MG637051 accession is now included.
- “351 ‘samples collected in Hungary.’”
Answer: Corrected.
- “383 ‘RDP v4.98 detected six distinct’ (delete ‘have’, spell out numbers up to ten unless followed by a unit of measure)”
Answer: Corrected to: “have predicted six” taking into account the opinion of reviewer 2.
- “392 ‘suggested to be playing a role in’”
Answer: Corrected.
- “420-421 ‘isolates clearly form a distinct clade’”
Answer: Corrected.
- “422 ‘Within this monophyletic group’”
Answer: Corrected.
- “427 ‘tulips and lilies each also form a separate’”
Answer: Corrected.
- “460 ‘amaranticolor’ (delete the ‘h’)”
Answer: Corrected.
- “461 ‘Plates were read at’”
Answer: Corrected.
- “462 ‘wavelength on a Labsystems Multiskan’”
Answer: Corrected.
- „l.500-502 Missing ‘MG637051’ – used at least for the phylogenetic network (Figure S3)”
Answer: The accession is now included.
- “553 ‘Alignment S7’ (not ‘S6’)”
Answer: Corrected.
Reviewer 2 Report
This manuscript describes the identification of several potyviruses associated with tulip color breaking syndrome in Hungary:
Lily mottle virus (LMoV), Tulip breaking virus (TBV) and Rembrandt tulip-breaking virus (ReTBV) were identified. these were sequenced and recombination and phylogenetic analyses performed. Following results obtained, ReTBV is proposed to be elevated to the species level.
The manuscript is well written and clear.
My only criticism is that the RDP program is only able PREDICT recombination events, to identify POTENTIAL recombination events. This should be clear when reading the text.
minor comments:
line 104: "The potyvirus species were depicted on map, to analyze the pattern of distribution" I do not understand what is ment in this sentence
line 105: LMoV and TBV were present IN diverse locations ...
line 392: Previously recombination was suggested to be playing A role in some potyvirus evolution towards host specialization ....
Author Response
Response to Editors and Reviewers’ Comments
Manuscript ID: plants-1035725
Title: Genetic diversity of potyviruses associated with tulip breaking syndrome
Dear Ms. Juicy Yang,
We would like to thanks the Associate Editor and the Reviewers for the effort and time put into the review of our manuscript (MS).
We have carefully considered the comments and we provide responses all to them.
Sincerely yours,
Laszlo Palkovics
Comments from Reviewer #2:
- „ My only criticism is that the RDP program is only able PREDICT recombination events, to identify POTENTIAL recombination events. This should be clear when reading the text.”
Answer: We have corrected this through the MS.
- “minor comments:
line 104: "The potyvirus species were depicted on map, to analyze the pattern of distribution" I do not understand what is ment in this sentence”
Answer: We wanted to illustrate the spatial distribution of virus infected tulip plants. We have re-prased the sentence to be more clear.
- “line 105: LMoV and TBV were present IN diverse locations ...”
Answer: Corrected to AT, according to the suggestion of Reviewer 1.
- “line 392: Previously recombination was suggested to be playing A role in some potyvirus evolution towards host specialization ....”
Answer: Corrected.